# The Prognostic Role of Human Papillomavirus and p16 Status in Penile Squamous Cell Carcinoma—A Systematic Review

**DOI:** 10.3390/cancers15143713

**Published:** 2023-07-21

**Authors:** Kevin Parza, Arfa Mustasam, Filip Ionescu, Mahati Paravathaneni, Reagan Sandstrom, Houssein Safa, G. Daniel Grass, Peter A. Johnstone, Steven A. Eschrich, Juskaran Chadha, Niki Zacharias, Curtis A. Pettaway, Philippe E. Spiess, Jad Chahoud

**Affiliations:** 1Internal Medicine, USF Health Morsani College of Medicine, Tampa, FL 33601, USA; kparza@usf.edu (K.P.); filip.ionescu@moffitt.org (F.I.); 2Genitourinary Oncology Department, H. Lee Moffitt Cancer Center, Tampa, FL 33601, USA; arfa.mustasam@moffitt.org (A.M.); mahati.paravathaneni@moffitt.org (M.P.); steven.eschrich@moffitt.org (S.A.E.); juskaran.chadha@moffitt.org (J.C.); philippe.spiess@moffitt.org (P.E.S.); 3Graduate Medication Education, USF Health Morsani College of Medicine, Tampa, FL 33602, USA; reagansandstrom@usf.edu; 4Hematology Oncology Department, Baylor College of Medicine, Houston, TX 77001, USA; houssein.m.safa@gmail.com; 5Radiation Oncology Department, H. Lee Moffitt Cancer Center, Tampa, FL 33601, USA; daniel.grass@moffitt.org (G.D.G.); peter.johnstone@moffitt.org (P.A.J.); 6Department of Urology, M.D. Anderson Cancer Center, University of Texas, Houston, TX 77001, USA; nmzacharias@mdanderson.org (N.Z.); cpettawa@mdanderson.org (C.A.P.)

**Keywords:** penile cancer, human papillomavirus, p16, HPV-related cancers, rare cancers

## Abstract

**Simple Summary:**

Penile squamous cell carcinoma (PSCC) is a rare and aggressive cancer. About half of all PSCC cases are thought to be related to human papillomavirus (HPV) or have increased expression of p16 by immunohistochemistry (IHC), a surrogate marker for HPV infection. HPV/p16 positivity is generally a marker of better outcomes in more common squamous cell carcinomas, but the studies evaluating its prognostic value in PSCC are conflicting. In this systematic review, we aim to summarize the existing literature on the impact of these biomarkers on PSCC prognosis and determine their potential value in identifying patients with more favorable biology.

**Abstract:**

PSCC is a rare cancer, with approximately half of all cases related to HPV. While HPV and p16 IHC testing have proven their prognostic value for oropharyngeal cancer, this is not yet established for PSCC. The current level of evidence exploring the relation between PSCC and HPV is moderate, so we conducted a systematic review following PRISMA guidelines to evaluate the prognostic role of HPV and p16 IHC in PSCC clinical outcomes. We searched the PubMed, Embase, and Cochrane databases and identified 34 relevant studies that met our inclusion criteria. Of these, 33 were retrospective cohort studies, and one was a cross-sectional study. Nine studies reported that HPV-positive and p16-positive PSCC had better overall survival (OS) and disease-free survival (DFS). This study highlights the need for a meta-analysis to determine the role of routine HPV status or p16 staining testing as part of the initial diagnosis and staging of PSCC patients worldwide.

## 1. Introduction

Penile squamous cell carcinoma (PSCC) is a rare genitourinary cancer with fewer than 2500 newly diagnosed cases and 500 deaths annually reported in the United States [1]. Incidence and mortality are even higher in developing countries [2]. Although surgery can be curative in localized cases, lymph node involvement in locally advanced and advanced PSCC confers a significantly worse prognosis and typically warrants a multimodal therapeutic approach [3,4,5,6,7].

The development of PSCC is frequently linked to multiple risk factors, such as inadequate hygiene, absence of circumcision, tobacco smoking, specific genetic conditions, and the presence of HPV, which can be found in 30–50% of cases [8,9,10,11,12].

Two markers, namely p16 overexpression and HPV status, are associated with HPV in PSCC and play a crucial role in determining prognostic significance. Generally, PSCC cases that are positive for HPV and p16 exhibit a more favorable prognosis compared to those that are HPV-negative. Consequently, determining the HPV and p16 status can aid in estimating prognosis and making informed treatment decisions [8].

While p16-positive status through immunohistochemistry (IHC) is considered a robust marker of HPV infection in oropharyngeal squamous cell carcinoma, according to the American Society of Clinical Oncology (ASCO) and National Comprehensive Cancer Network (NCCN) guidelines, its value in PSCC is less clear [13,14]. Studies on the prognostic potential and predictive value of HPV and p16 in PSCC have yielded contradictory results [8,15]. Currently, testing HPV status in the initial diagnostic workup of PSCC carries a level 2a evidence grade in the updated ASCO–European Association of Urology (EAU) guidelines, which emphasize the need for more data to support the recommendation of universally assessing HPV status in all PSCC patients [16].

A previous systematic review and meta-analysis of twenty studies concluded that men with HPV-positive or p16-positive PSCC had significantly better disease-specific survival (DSS) [8]. However, the relationship between these biomarkers and overall survival (OS) or disease-free survival (DFS) was inconsistent among individual studies, and no statistically significant associations were found [8]. To complement this prior effort, we conducted an updated systematic review with a larger sample size and more widespread biomarker testing to explore the prognostic significance of HPV and p16 status on PSCC outcomes [8].

## 2. Materials and Methods

### 2.1. Search Strategy

We performed a systematic literature search in the electronic databases PubMed, Embase, and the Cochrane Library, covering articles published between March 1992 and December 2022. The database search was filtered to include articles written in English. The search terms used were “Prognosis” OR “Prevalence” AND “HPV” OR “Human Papillomavirus” OR “Human Papillomaviruses” OR “Papillomaviridae” OR “human papillomavirus” OR “Cyclin-Dependent Kinase Inhibitor p16” OR “p16INK4” OR “Cyclin-Dependent Kinase Inhibitor-2A” OR “p16INK4A Protein” OR “CDKN2A Protein” OR “CDKN2 Protein” AND “Penile Neo-plasms” OR “Penile Neoplasms” OR “Penis Neoplasms” OR “Penis Neoplasm” OR “Penile Neoplasm” OR “Penis Cancer” OR “Penile Cancer” OR “Penile carcinoma” OR “penis carcinoma” OR “penis tumor” OR “penile cancers”.

JC conducted the initial search, and subsequently, KP and AM independently verified the abstracts and selected manuscripts. Disagreements were resolved through collaborative discussions to achieve consensus. A fourth author, RS, addressed any unresolved conflicts. Supplementary information was reviewed using reference lists to ensure comprehensiveness. The study adhered to the guidelines outlined by the Preferred Reporting Items for Systematic Reviews and Meta-Analyses (PRISMA) checklist (Figure 1) [17].

### 2.2. Study Selection

Only peer-reviewed studies in English or with available translations were eligible for inclusion in the analysis. In cases where a study population was reported in multiple papers, only data from the most up-to-date paper and providing the most complete information were utilized. Case reports, abstracts, letters to the editor, and review articles were excluded from the analysis.

### 2.3. Quality Assessment

Two reviewers independently evaluated the quality of the chosen papers using a modified Newcastle–Ottawa Scale (NOS). The scale comprised 8 items divided into three domains: subject selection criteria (0–4 points), comparability of subjects (0–2 points), and outcomes (0–3 points). The maximum achievable score was 9 points, with a score of more than 5 points indicating high quality. Appendix A provides a summary of the assessment of the NOS scores with a median total score of 7 (IQR 7–8.25).

### 2.4. Data Extraction

A total of 34 articles were identified and selected for data extraction. Data tables were created to systematically capture relevant information from the texts, tables, and figures of each included study. These tables included details such as the first author, publication year, country, year of sample collection, follow-up time, type of tissue testing method, p16 testing technique, the definition of p16 positivity (overexpression), number of evaluators for p16 staining, sample size, the number of HPV-positive and HPV-negative PSCC cases, the number of p16-positive PSCC cases, and various survival endpoints. The examined survival outcomes encompassed overall survival (OS), disease-specific survival (DSS), and disease-free survival (DFS).

### 2.5. Mean Age Adjustment

During our research, we encountered studies that presented the median age of patients, along with either the interquartile range (IQR) or range, rather than providing the mean and standard deviation. To approximate the mean age, we adopted the methodology introduced by Wan et al. [18] in 2014, which provides formulas to estimate the sample mean and standard deviation based on factors such as the sample size, median, and either the IQR or range [18]. By employing these formulas, we successfully estimated the mean age of patients in the aforementioned studies and calculated the overall mean for our sample size.

## 3. Results

### 3.1. Study Characteristics

In this systematic review, we conducted an analysis of the data derived from a total of 34 studies encompassing 3994 patients diagnosed with PSCC. These studies spanned the period from 1970 to 2022. The estimated mean age of the patients was determined to be 62.3 years. Among the studies included in our analysis, Brazil and the United States both had the highest number of studies, with each country contributing eight studies (refer to Figure 2 for details). Fourteen studies reported on both HPV and p16 IHC status [4,13,19,20,21,22,23,24,25,26,27,28,29,30], fifteen studies [15,31,32,33,34,35,36,37,38,39,40,41,42,43,44] focused solely on HPV, and five studies [45,46,47,48,49] reported only on p16 status. For a summary of HPV and p16 positivity across the 34 included studies, please refer to Table 1 and Appendix A.

### 3.2. HPV Detection Methods

Among the studies included in our analysis, a total of twenty-nine studies reported HPV status. Out of these, eighteen studies employed the polymerase chain reaction (PCR) method for HPV detection [13,15,19,20,22,23,25,26,27,28,31,32,33,35,36,37,38,39], six studies utilized in situ hybridization (ISH) [4,21,24,29,34,40], and four studies did not specify the method used for HPV detection [41,42,43,44]. Additionally, one study employed a Quantus fluorometer for the detection of HPV nucleic acid [30]. For further details, please refer to Appendix A.

### 3.3. p16 Detection Methods

Nineteen studies included in our analysis examined samples to evaluate the p16+ status using immunohistochemical (IHC) staining. Two different methodologies were employed for grading p16 on IHC. Six studies utilized a quantitative measurement approach, where the cut-off for considering a sample as p16-positive ranged from >10% to >75% of cells exhibiting positive staining [4,20,24,26,29,48]. Conversely, twelve studies employed subjective criteria to define p16 positivity, using descriptions such as strong and diffuse staining, vital staining of proliferative cells, continuous and complete cytoplasmic staining, intense confluent staining, or focal scattering of staining in the cytoplasm and nucleus of cells [13,19,21,22,23,25,28,30,42,45,47,49]. Bethune et al. [46] adopted a mixed methodology, considering a qualitative cut-off of >30% staining with nuclear and cytoplasmic staining, moderate sample staining with at least 30% of cells staining, or strong cytoplasmic and nuclear staining as positive for p16 [46]. One study did not specify the criteria used to determine p16 positivity [27]. For further details, please refer to Appendix A.

### 3.4. HPV + and p16+ Impact on Overall Survival

None of the included studies demonstrated a significant association between HPV positivity and improved overall survival (OS). However, p16 status was found to be associated with improved OS in three independent analyses [4,29,46]. Notably, Chahoud et al. [4] investigated both biomarkers in the same population and discovered a positive correlation between p16-positive cases and superior OS, while no such association was observed for HPV+ cases [4]. These discrepancies could potentially be attributed to the variations in defining p16 positivity during IHC analysis, indicating the presence of heterogeneity among the studies. For further details on the impact HPV and p16 positivity had on overall survival, please refer to Table 2.

### 3.5. HPV+ and p16+ Impact on Disease-Free Survival

The presence of HPV positivity was associated with improved disease-free survival (DFS) and was statistically significant in four of the studies [13,26,28,39]. Similarly, p16 positivity was also associated with improved DFS and was statistically significant in two of the studies [42,47]. For further details on the impact HPV and p16 positivity had on disease-free survival, please refer to Table 3.

### 3.6. HPV+ and p16+ Impact on Disease-Specific Survival

In three of the studies, HPV positivity was associated with improved disease-specific survival (DSS) [28,32,37]. Conversely, p16 positivity was linked with improved DSS in four of the studies [4,24,28,45]. Notably, the study conducted by Chu et al. [28] demonstrated statistically significant associations for both biomarkers within the same population [28]. For further details on the impact HPV and p16 positivity had on disease-specific survival, please refer to Table 4.

## 4. Discussion

This systematic review presents an analysis of 34 studies, aiming to provide an up-to-date overview of the prognostic significance of HPV status and p16 expression in PSCC. The inclusion of recent studies has significantly enhanced previous efforts on this topic, notably by substantially increasing the sample size and incorporating a more diverse patient population from various global sites.

The risk factors for PSCC can vary across countries and regions. While some risk factors, like HPV, are consistent globally, others may be more prevalent in specific populations. For example, Brazil has been identified as having one of the highest incidence rates of PSCC worldwide, possibly due to a high prevalence of HPV infection, lower rates of circumcision, and socioeconomic factors such as limited access to healthcare and education about safe sex practices [11]. On the other hand, westernized countries like Portugal may have a lower incidence of PSCC due to public health campaigns and educational initiatives that raise awareness about the disease and promote preventive measures such as good hygiene and regular medical check-ups. These efforts may facilitate timely diagnosis and treatment, potentially reducing the occurrence of PSCC [50].

Certain high-risk HPV genotypes, particularly HPV-16 and HPV-18, are strongly associated with p16 overexpression in infected cells. HPV-16—being the most oncogenic and prevalent high-risk genotype—is strongly linked to the development of various cancers, including cervical, anal, vulvar, vaginal, certain head and neck cancers, and penile cancers [51,52]. In contrast, low-risk HPV genotypes like HPV-6 and HPV-11 are commonly associated with benign lesions and typically not associated with p16 overexpression [53,54].

It is crucial to note that combining multiple prevention strategies, including HPV vaccination, screening, and safe sexual practices, can have the greatest impact on reducing HPV-related diseases. However, challenges such as vaccine hesitancy, lack of awareness, and limited access to vaccination programs can hinder vaccine uptake. Sociocultural factors and stigma surrounding discussions about sexual health and genital cancers may also create barriers to open dialogue, education, and prevention efforts [55,56,57].

The 2022 World Health Organization (WHO) classification of penile cancer, as suggested by Menon et al., recommends the utilization of p16 screening whenever possible to differentiate between HPV-associated and HPV-independent PSCC [58]. This updated recommendation is expected to result in increased p16 testing and the accumulation of additional studies, which can contribute to the statistical power of future meta-analyses.

In our review, we conducted a multivariate analysis and found that p16 positivity was associated with improved OS and DSS in a subset of published studies. Specifically, in 3 out of 11 (27%) studies, p16 positivity was associated with improved OS, and in 4 out of 7 (57%) studies, p16 positivity was associated with improved DSS. In comparison, HPV positivity was not significantly associated with improved OS in any of the 10 studies analyzed (0%), and it was associated with improved DSS in only 3 out of 10 (30%) studies. Most of the included studies in our review demonstrated good methodological quality, with a majority achieving a Newcastle–Ottawa Scale (NOS) score higher than 7 and a median total score of 7 (IQR 7–8.25). However, it is important to acknowledge the limitations of our systematic review. We did not include unpublished studies or studies published in languages other than English, which may introduce publication bias and limit the comprehensiveness of the evidence base. Furthermore, another limitation in our study arises from the observed variability in the definitions of p16 positivity among the twenty included studies that assessed p16 status. Out of these studies, six [4,20,24,26,29,48] employed a quantitative measurement approach, while twelve [13,19,21,22,23,25,28,30,42,45,47,49] relied on descriptive criteria. Additionally, Bethune et al. [46] and Chahoud et al. [4] utilized both methodologies, reporting both quantitative and descriptive elements to define p16 positivity. The existence of such diverse definitions for p16 positivity presents a challenge in drawing definitive conclusions and synthesizing the findings across studies.

The inclusion of fourteen additional studies, since the publication of the meta-analysis by Sand et al. [8], provides significant additions to the current body of evidence regarding the prognostic significance of HPV and p16 biomarkers in PSCC. Although the systematic review has not yielded conclusive results regarding the overall data on HPV and P16 biomarkers, it has provided new evidence of the possible correlation of p16 using IHC with better prognosis in PSCC and highlighted some discrepancy between p16 and HPV status in PSCC.

The current guidelines from EAU–ASCO recommend conducting p16 testing through IHC in all individuals diagnosed with PSCC [16]. This underscores the importance of assessing p16 expression as a valuable diagnostic tool in PSCC management. Looking ahead, we anticipate that an upcoming meta-analysis will provide additional clarity and a deeper understanding of this subject in the future, further advancing our knowledge of the prognostic significance of HPV and p16 biomarkers in PSCC.

## 5. Conclusions

Our systematic review highlights the importance of conducting an updated comprehensive meta-analysis to define the potential role of these biomarkers in PSCC. We anticipate that this will address the discordance between p16 and HPV and uncover a potentially stronger prognostic value for p16 positivity on the outcomes of PSCC patients.

## Figures and Tables

**Figure 1 cancers-15-03713-f001:**
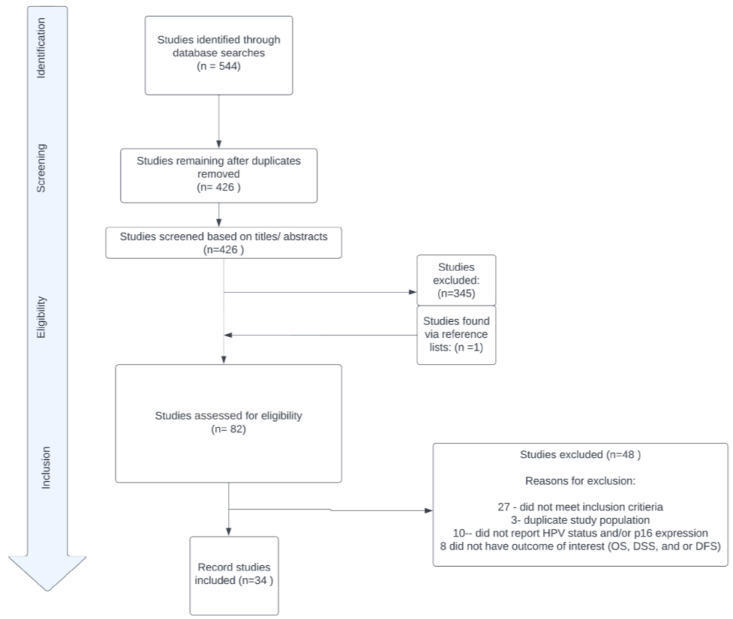
PRISMA diagram.

**Figure 2 cancers-15-03713-f002:**
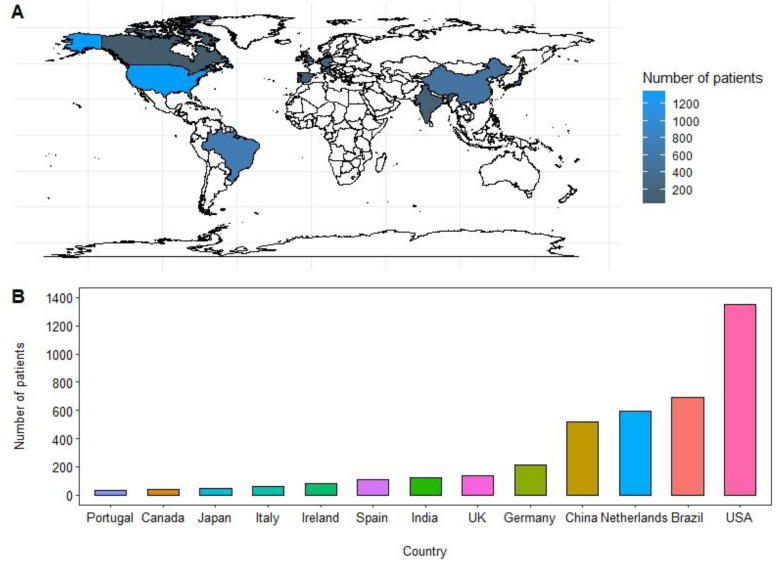
(**A**,**B**) Original geographical distribution and bar graph representation of number of patients included in the study from 1970–2022.

**Table 1 cancers-15-03713-t001:** Prevalence of HPV and p16 positivity across the included studies.

Study ID	HPV Status	p16 Status
Wiener et al., 1992 [31]	29 Tested 9 Positive 31% Positive	N/A
(Artur) Bezerra et al., 2001 [15]	82 Tested 25 Positive 30% Positive	N/A
Lont et al., 2006 [32]	171 Tested 50 Positive 29% Positive	N/A
Guerrero et al., 2008 [19]	24 Tested 11 Positive 46% Positive	# Tested = 24 # Positive = 15 % Positive = 63%
Scheiner et al., 2008 [33]	72 Tested 58 Positive 81% Positive	N/A
Ferrandiz-Pulido et al., 2013 [20]	77 Tested 31 Positive 40% Positive	67 Tested 23 Positive 34% Positive
Gunia et al., 2012 [45]	N/A	92 Tested 54 Positive 59% Positive
Bethune et al., 2012 [46]	N/A	40 Tested 23 Positive 58% Positive
Dilorenzo et al., 2013 [34]	30 Tested 8 Positive 26% Positive	N/A
de Fonseca et al., 2013 [35]	82 Tested 50 Positive 61% Positive	N/A
Hernandez et al., 2014 [36]	79 Tested 50 Positive 63% Positive	N/A
(Stephania) Bezerra et al., 2015 [21]	53 Tested 8 Positive 15% Positive	52 Tested 23 Positive 44% Positive
Djajadiningrat et al., 2015 [37]	212 Tested 53 Positive 25% Positive	N/A
McDaniel et al., 2015 [22]	43 Tested 5 Positive 12% Positive	38 Tested 11 Positive 28% Positive
Steinestel et al., 2015 [23]	58 Tested 18 Positive 31% Positive	58 Tested 34 Positive 59% Positive
Tang et al., 2015 [47]	N/A	119 Tested 59 Positive 50% Positive
Zargar-Shoshtari et al., 2016 [24]	57 Tested 24 Positive 42% Positive	57 Tested 23 Positive 40% Positive
Afonso et al., 2017 [25]	122 Tested 79 Positive 65% Positive	99 Tested 28 Positive 28% Positive
de Araújo et al., 2018 [38]	179 Tested 56 Positive 31% Positive	N/A
Ottenhof et al., 2018 [39]	213 Tested 52 Positive 24% Positive	N/A
Vicenilma Martins et al., 2018 [13]	55 Tested 49 Positive 89% Positive	55 Tested 22 Positive 40% Positive
Takamoto et al., 2018 [40]	44 Tested 5 Positive 11% Positive	N/A
De Bacco et al., 2020 [48]	N/A	35 Tested 13 Positive 37% Positive
Wang et al., 2020 [41]	292 Tested 130 Positive 45% Positive	N/A
Ashley et al., 2020 [42]	137 Tested 74 Positive 54% Positive	N/A
Pereira-Lourenço et al., 2020 [49]	N/A	35 Tested 8 Positive 23% Positive
Valquíria Martins et al., 2020 [26]	47 Tested 21 Positive 45% Positive	26 Tested 12 Positive 46% Positive
Muresu et al., 2020 [27]	32 Tested 9 Positive 28% Positive	32 Tested 7 Positive 22% Positive
Chu et al., 2020 [28]	226 Tested 74 Positive 33% Positive	226 Tested 59 Positive 26% Positive
Chipollini et al., 2021 [43]	825 Tested 321 Positive 39% Positive	N/A
Mohanty et al., 2021 [29]	123 Tested 57 Positive 46% Positive	123 Tested 65 Positive 53% Positive
Müller et al., 2020 [27]	58 Tested 33 Positive 57% Positive	60 Tested 31 Positive 52% Positive
Browne et al., 2022 [44]	81 Tested 45 Positive 56% Positive	N/A
Chahoud et al., 2022 [4]	143 Tested 47 Positive 33% Positive	143 Tested 45 Positive 32% Positive
Median (%positive)	79 (39%)	57(44%)
Q1	53 (29%)	37 (33%)
Q3	143 (54%)	94 (53%)

**Table 2 cancers-15-03713-t002:** HPV+ and p16+ on overall survival (OS).

Study ID	HPV+ HR with 95% CI			p16+ HR with 95% CI		
	HR	Lower Limit	Upper Limit	HR	Lower Limit	Upper Limit
Wiener et al. 1992 [31]	1.15	0.39	3.36			
Guerrero et al. 2008 [19]	4.37	0.51	37.47	0.56	0.11	2.83
Ferrandiz-pulido et al. 2013 [20]	0.28	0.061343	1.278	0.28	0.061343	1.278
(Steph) Bezerra et al. 2015 [21]	1.76	0.61	5.12	1.53	0.52	4.54
Hernandez et al. 2014 [36]	0.86	0.4	1.86			
Chippollini et al. 2021 [43]	0.89	0.67	1.19			
Martins et al. 2020 [26]	0.92471	0.37166	2.3	0.7067	0.23	2.164
Chahoud et al. 2022 [4]	0.8362	0.497	1.4068	0.36 *	0.2	0.67
Muresu et al. 2020 [27]	0.6	0.1	2.7	1	0.2	5.2
De Bacco et al. 2019 [48]				1.52	0.5	4.62
Muller et al. 2021 [30]	0.607	0.201	1.83	0.628	0.207	1.899
Bethune et al. 2012 [46]				0.55 *	0.31	0.94
Mohanty et al. 2021 [29]				0.32 *	0.2	0.5
Brown et al. 2022 [44]				0.461	0.1719	1.236

Note: (*) denotes statistically significant data.

**Table 3 cancers-15-03713-t003:** HPV+ and p16 + on disease-free survival (DFS).

Study ID	HPV+ HR with 95% CI			p16+ HR with 95% CI		
	HR	Lower Limit	Upper Limit	HR	Lower Limit	Upper Limit
Guerrero et al. 2008 [19]	1.02	0.31	3.37	0.91	0.29	2.91
Scheiner et al. 2008 [33]	0.69	0.33	1.46			
Dilorenzo et al. 2013 [34]	0.8	0.1	4.2			
de Fonseca et al. 2013 [35]	0.87	0.49	1.55			
Afonso et al. 2017 [25]	0.8056	0.377	1.72			
Martins et al. 2020 [26]	0.3 *	0.09	0.97			
Chu et al. 2020 [28]	0.334 *	0.158	0.705			
Martins et al. 2018 [13]	0.1234 *	0.0407	0.373			
Ottenhof et al. 2018 [39]	0.207 *	0.049	0.87			
de Araujo et al. 2018 [38]	0.8134	0.365	1.8078			
Tang et al. 2015 [47]				0.066 *	0.007	0.556
Ashley et al. 2020 [42]				0.41 *	0.18	0.94
Per Lourenco et al. 2020 [49]				0.55	0.11	2.57
Browne et al. 2022 [44]				0.7403	0.281	1.946

Note: (*) denotes statistically significant data.

**Table 4 cancers-15-03713-t004:** HPV+ and p16+ on disease-specific survival (DSS).

Study ID	HPV+ HR with 95% CI			p16+ HR with 95% CI		
	HR	Lower Limit	Upper Limit	HR	Lower Limit	Upper Limit
Wiener et al. 1992 [31]	1.37	0.43	4.32			
(Artur) Bezerra et al. 2001 [15]	0.96	0.41	2.27			
Lont et al. 2006 [32]	0.21 *	0.06	0.76			
Ferrandiz-pulido et al. 2013 [20]	0.26	0.04	1.91			
(Steph) Bezerra et al. 2015 [21]	1.05	0.21	5.07	0.54	0.13	2.19
Djajadiningrat et al. 2015 [37]	0.2 *	0.1	0.9			
Steinestel et al. 2015 [23]	0.14	0.02	1.04	0.51	0.2	1.3
Chu et al. 2020 [28]	0.38 *	0.18	0.82	0.56 *	0.183	0.824
Chahoud et al. 2022 [4]	0.5964	0.293	1.211	0.2 *	0.07	0.54
Takamoto et al. 2017 [40]	0.7416	0.094	5.7955			
Gunia et al. 2011 [45]				0.25 *	0.08	0.79
Bethune et al. 2012 [46]				0.53	0.26	1.06
Zargar-Shoshtari et al. 2016 [24]				0.36 *	0.13	0.99

Note: (*) denotes statistically significant data.

## Data Availability

The data presented in this study are available on request from the corresponding author.

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
