# Peer review of "The Prognostic Role of Human Papillomavirus and p16 Status in Penile Squamous Cell Carcinoma—A Systematic Review"

_cancers, 2023, doi:10.3390/cancers15143713_

Round 1

Reviewer 1 Report

Dear authors

The manuscript is interesting and well-written. There are some comments with this regard.

1-      The description of PSCC status and risk factors in the introduction is needed.

2-      In the discussion section, it is proposed to include major risk factors and discuss in various countries.

3-      The control and prevention strategies can be added and challenges in various age ranges is warranted.

4-      Please add a description regarding HPV genotypes and their carcinogenic roles and p16 protein status among HPV genotypes.  

5-      You can use more papers in 2023 such as these recently published papers:

Metallic Nanoparticles: Their Potential Role in Breast Cancer Immunotherapy via Trained Immunity Provocation. Biomedicines. 2023

p16 status and high-risk human papillomavirus infection in squamous cell carcinoma of the nasal vestibule, 2023 

Kind regards  

Reviewer 2 Report

The authors summarize the evidence in the literatura of biomarkers on PSCC prognosis, introduction, metholodogy and results are appropiate. Please include the results of  the metha-analysis of the 34 studies  presented analysed.

Author Response

Comments and Suggestions for Authors

The authors summarize the evidence in the literature of biomarkers on PSCC prognosis, introduction, methodology and results are appropriate. Please include the results of  the meta-analysis of the 34 studies  presented analyzed.

Thank you for your kind review and comments, the scope of this submission was a systematic review only, future work will focus on completing a meta-analysis.

Reviewer 3 Report

The review explores the relationship between HPV/p16 positivity and patient outcomes, which is a valuable effort since conflicting data exists, whereas in other HPV related cancers, HPV positivity clearly has better outcomes than HPV-.

The manuscript is well written.

my only suggestion is to incorporate a short paragraph in the discussion section, relating the findings of this review with other cancers, to illustrate how different the results e.g. in head / neck and gyn are. Also, a sentence at the end stating that the new eau/asco guideline now clearly recommend P16 testing in ALL penile cancer patients which is expected to help elucidate this topic further in the future.

Author Response

Please the attachment. Thanks so much!
